# Icephobic and Anticorrosion Coatings Deposited by Electrospinning on Aluminum Alloys for Aerospace Applications

**DOI:** 10.3390/polym13234164

**Published:** 2021-11-28

**Authors:** Adrián Vicente, Pedro J. Rivero, Paloma García, Julio Mora, Francisco Carreño, José F. Palacio, Rafael Rodríguez

**Affiliations:** 1Engineering Department, Campus de Arrosadía S/N, Public University of Navarre, 31006 Pamplona, Spain; pedrojose.rivero@unavarra.es (P.J.R.); rafael.rodriguez@unavarra.es (R.R.); 2Institute for Advanced Materials and Mathematics (INAMAT^2^), Campus de Arrosadía S/N, Public University of Navarre, 31006 Pamplona, Spain; 3INTA-Instituto Nacional de Técnica Aeroespacial, Área de Materiales Metálicos, Ctra. Ajalvir Km 4, 28850 Torrejón de Ardoz, Spain; garciagp@inta.es (P.G.); jmornog@inta.es (J.M.); fcarpue@inta.es (F.C.); 4Centre of Advanced Surface Engineering, AIN, 31191 Cordovilla, Spain; jfpalacio@ain.es

**Keywords:** electrospinning, PVDF-HFP, corrosion resistance, super hydrophobic, SLIPS, ice adhesion

## Abstract

Anti-icing or passive strategies have undergone a remarkable growth in importance as a complement for the de-icing approaches or active methods. As a result, many efforts for developing icephobic surfaces have been mostly dedicated to apply superhydrophobic coatings. Recently, a different type of ice-repellent structure based on slippery liquid-infused porous surfaces (SLIPS) has attracted increasing attention for being a simple and effective passive ice protection in a wide range of application areas, especially for the prevention of ice formation on aircrafts. In this work, the electrospinning technique has been used for the deposition of PVDF-HFP coatings on samples of the aeronautical alloy AA7075 by using a thickness control system based on the identification of the proper combination of process parameters such as the flow rate and applied voltage. In addition, the influence of the experimental conditions on the nanofiber properties is evaluated in terms of surface morphology, wettability, corrosion resistance, and optical transmittance. The experimental results showed an improvement in the micro/nanoscale structure, which optimizes the superhydrophobic and anticorrosive behavior due to the air trapped inside the nanotextured surface. In addition, once the best coating was selected, centrifugal ice adhesion tests (CAT) were carried out for two types of icing conditions (glaze and rime) simulated in an ice wind tunnel (IWT) on both as-deposited and liquid-infused coatings (SLIPs). The liquid-infused coatings showed a low water adhesion (low contact angle hysteresis) and low ice adhesion strength, reducing the ice adhesion four times with respect to PTFE (a well-known low-ice-adhesion material used as a reference).

## 1. Introduction

Ice accretion on surfaces presents a serious challenge from an operational, economical, and safety perspective especially in cold climates. Some of the most remarkable examples of icing problems are on aircraft [1], wind turbines [2], transmission lines, and communication towers [3]. Therefore, the development of anti-icing and ice-release coatings has been a subject of great interest over the last decades.

Current mechanisms for ice removal comprise two different strategies: de-icing and anti-icing.

The de-icing approach (active method) involves chemical, thermoelectric, or mechanical methods for removing ice that has already accumulated or, when previously activated, for helping to avoid the ice accretion. Therefore, in order to implement light, efficient, and compact de-icing active systems, the anti-icing strategies play an important role as a complement for the de-icing approaches. These anti-icing methods (passive methods) are based on the icephobicity of the surfaces to prevent the accretion, or facilitate the shedding. In recent years, anti-icing coatings research has experienced considerable progress by mimicking nature’s materials. For example, the polar bears, which live in the world’s coldest environments, have a special hair that prevent the penetration of cold sea water to the skin, and their corresponding fur exhibits excellent superhydrophobicity and antiadhesion characteristics needed to repel water microdroplets [4]. In addition, polar bear fur is oily; as a result, an increment of the water repellency is generated. Thus, they can easily shake free of water and any ice that may form after swimming.

The wettability of a solid surface can be defined in terms of water contact angle (WCA) [5]. Normally, a hydrophobic surface shows partial wettability and high WCA (θ > 90°), while a superhydrophobic surface completely repels water droplets because of zero wettability and very high WCA (θ > 150°) [6]. In these superhydrophobic surfaces (SHS), the resultant deposited water droplets can remain in a nonwetting Cassie–Baxter state [7], resulting from low-surface-energy compounds and air trapping inside the textured surface [8,9,10]. The trapped air minimizes the interaction between the water droplet and solid surfaces [11]. Thus, the energy barrier to the removal of water droplets from SHS is decreased. Consequently a wide variety of industrial applications such as anti-icing [12], antibacterial [13], self-cleaning [14] or even corrosion-resistant surfaces [15] have been developed.

On account of the deficiency of SHS, the SLIPS was inspired by the insect-trapping mechanism of pitcher plants [16]. SLIPS consists of rough-textured (porous), solid substrate impregnated with a lubricant, typically an oil, that forms a homogeneous smooth defect-free interface. It could be considered an analogous to SHS where the air trapped between substrate and water is substituted by the lubricant. The lubricant-impregnated surface has shown superior nonwetting performance, excellent liquid-repellency, pressure stability, high tolerance of humidity, and optical transparency [17]. The infused oil viscosity plays an important part in the motion of tested droplets, whether they roll or slip.

In addition, the surface with lubricating oil locked in the holes of the micro/nanoscale substrate [17,18,19] exhibits extreme liquid repellence, characterized by low contact angle hysteresis, low tilt angle, and long-term stable performance even under harsh environmental conditions. Therefore, SLIPS plays an interesting role in preventing undesired ice accretion phenomena and corrosion side effects for aeronautical applications, where atmospheric icing originating from supercooled droplets can cause serious problems to the aircrafts, such as drift and drag, as well as catastrophic imbalances at high flying speed [20,21,22,23,24,25].

Up to now, different fabrication methods have been used for the production of porous surfaces for icephobic SLIPS, such as laser writing [26,27], self-assembly [28,29], phase separation [30] and electrospraying [31]. However, a simple, cost-effective, scalable and highly versatile method for the large-scale fabrication of superhydrophobic micro/nanofibers is electrospinning [32,33,34]. This technique can provide a low surface energy as a function of the selected polymeric precursor [35] as well as a highly rough structure caused by the hierarchical microstructures and nanostructures of micro/nanofibrous mat [36], which could show a similar effect to that of polar bear fur [4]. The morphology surface roughness and the structure of the electrospun mats depend on the intrinsic polymeric precursor properties (concentration, molecular weight, viscosity, surface tension, and nature of solvent), the operational condition parameters (applied voltage, flow rate, and distance of tip to collector), and the environmental conditions (relative humidity and temperature) [37,38,39]. 

In this study, a nanofibrous coating of PVDF-HFP prepared using electrospinning was employed to obtain a porous surface for icephobic SLIPS. A proper deposition mode has been developed by controlling the flow rate and applied voltage. In addition, a thickness control system was implemented to obtain electrospun coating with 200 µm of thickness. As a result, an accurate identification of the electrospun samples in terms of surface morphology, wettability, corrosion resistance, and UV-VIS-NIR optical properties has been obtained, applying different input parameters in the electrospinning process. Moreover, the most representative factors, which directly affect the surface morphology (fiber diameter, fiber density, the fiber size variability, and surface roughness) have been compared with the wettability, corrosion resistance, and optical transmittance/reflectance to find the different relationships, which define an optimal performance. Finally, the SLIPS coating was deposited and compared with the SHS, in order to analyze wettability (water adhesion strength) and ice adhesion performance.

## 2. Experimental Procedure

### 2.1. Materials and Reagents

AA7075-T6 aluminum alloy flat samples (AL-USTOCK, Vitoria, Spain) with a final dimension of 75 mm of length, 50 mm of width, and 2 mm of thickness were polished with a roughness less than (Sa = 0.8 µm; Ra = 0.7 µm) and cleaned with acetone. This specific aluminum alloy has been selected as the reference substrate to test the corrosion resistance due to it being one of the most important alloys used in aeronautical applications where high mechanical properties and corrosion resistance are need [40,41]. 

Poly(vinylidene fluoride-co-hexafluoropropylene) (PVDF-HFP, (–CH_2_CF_2_–)_x_[–CF_2_CF(CF_3_)–]_y_, Mw = 400,000 g/mol), dimethylformamide (DMF), acetone, and lubricant silicon oil (Grade: for oil baths from −50 °C to +200 °C, [–Si(CH_3_)_2_O–]_n_) were purchased from Sigma-Aldrich (Saint Luis, MO, USA). All reagents were used without any further purification. 

A first series of coating deposition was carried out on standard glass slides (76 × 52 mm^2^, Sigma-Aldrich) to identify the characteristics of the thickness, morphology, wettability, and optical properties. 

For all the other tests, the coatings were (a) directly deposited onto aluminum alloy substrates and (b) onto a aluminum alloy samples with a fresh layer of sprayed glue (SUPERGEN spray mounts glue, Ceys) intended to increase the adhesion of the coating to the substrate [42].

### 2.2. Electrospinning Procedure

PVDF-HFP was dissolved in a 45.5:54.5 (V_DMF_:V_AC_) solvents mixture of DMF and acetone (AC). Thus, a homogeneous PVDF-HFP solution of 15 wt% was obtained at room temperature and under stirring (200 rpm) for 12 h. 

The solution was loaded into a 10 mL syringe located vertically on a syringe pump, whereas the distance between the capillary tip and the aluminum flat collector was 15 cm. The positive electrode for all the depositions was a 20-gauge needle with an inner diameter of 0.6 mm. The electrospun samples were coated at room temperature (20 °C), at 40% humidity (RH), and at different deposition periods for a fixed thickness control of 200 µm.

In addition, some of the samples (denoted as SxG) were sprayed with a glue as a binder before the electrospinning process, to be distinguished from the rest (denoted as S_X_) without the sprayed glue layer, in order to understand the effect of the glue layer over the electrospun coating in terms of corrosion performance. Finally, the electrospun coated samples were immersed into the lubricant silicon oil for 1 h, and afterwards, all samples were kept in a 45° tilted plate overnight to get rid of excess oil, obtaining slippery liquid porous surfaces.

### 2.3. Characterization Techniques

First, the Fourier-transform infrared (FTIR) spectroscopy study was carried out using a Perkin Elmer spectrophotometer (Waltham, MA, USA) in the spectral range of 600–2000 cm^−1^ with the aim to show the functional groups of the electrospun fibers.

Then, on the one hand, the thickness control process of the electrospun PVDF-HFP coatings was carried out by measuring the thickness of a vertical profile image, using a confocal microscope (model S-mart, SENSOFAR METROLOGY, Barcelona, Spain). The confocal microscope applies a Corse Shift single algorithm with an objective of EPI 50X v35 and EPI 20X for an area of 340.03 × 283.73 μm^2^ and 850.05 × 709.32 µm^2^, respectively. On the other hand, the resultant surface morphologies of the electrospun samples have been measured, applying a Corse Shift single algorithm with an objective of EPI 50X v35 for an area of 340.03 × 283.73 μm^2^. According to the standard ISO 25178, the roughness surface (S-L) measurements have been obtained with three different filters: a low filter (F-operator-level, plane), a high filter (S-filter, standard cut off λs: 0.8 µm), and a Gaussian filter (L-filter, standard cut off λc: 0.08 mm, Sa < 0.02 µm). These topography measurements were used for estimating the average fiber diameter (Df) and the surface roughness (Sa) of the samples.

In addition, the surface morphologies of the electrospun samples were also measured by Atomic Force Microscope (AFM, Veeco Innova AFM, Veeco Instruments, Plainview, NY, USA) and field-emission scanning electron microscopy (FE-SEM, Hitachi S4800, Tokyo, Japan). On one side, the Gwyddion software was used to analyze the AFM images through a Flatten Base function to perform the levelling and the statistical Quantities Tool with the moment-based mode to obtain the mean roughness (Sa) value. On the other side, the ImageJ software was applied to measure the FE-SEM images to obtain the average fiber diameter (Df) of the samples.

Water contact angle (WCA) measurements were carried out with a CAM 100 contact angle goniometer (CAM 100, KSV Instruments, Burlington, VT, USA) using distilled water. The static water contact angle was measured 3 times and in 6 different places of the sample. Moreover, the sliding water angle ($\alpha_{slide}$) and the contact angle of hysteresis (CAH) were measured with an optical tensiometer (Attension Theta Lite, Biolin scientific, Gothenburg, Sweden) and with the module tilting stage C218.

Electrochemical measurements of Tafel polarization curves were carried out on an Autolab Potentiostat/Galvanostat PGSTAT302N (Metrohm, Herisau, Switzerland). All corrosion tests were performed at room temperature in 6 wt% NaCl aqueous solutions, using a conventional three-electrode cell consisting of a working electrode (bare or coated Al sample), a silver chloride Ag–AgCl reference electrode, and a platinum counter electrode. Before conducting all the experiments, the samples were immersed in the 6 wt% NaCl electrolyte for 30 min to make sure that the system was in steady state and with the open circuit potential (OCP) stabilized. The set-up employed for this measurement can be seen in Figure 1 of Reference [33].

Tafel polarization measurements were obtained by scanning the electrode potential automatically from −200 mV to +200 mV with respect to the OCP voltage at a scan rate of 1.5 mV s^−1^.

The output from these experiments yielded a polarization curve of the current density versus the applied potential. The resulting corrosion current can be calculated by using Tafel slope analysis, where it established a relationship between the current density and the electrode potential during the polarization. The corrosion data were obtained from Tafel polarization curves by superimposing a straight line on the linear portions of the cathodic and anodic curves. Finally, other corrosion parameters, such as equivalent weight of the metal, density, or exposed surface, are also required as input parameters. With this information, the software generates the complete set of corrosion parameters. Thus, the corrosion rate was calculated according to the following equation [43].
(1)$$Corrosion\;Rate=327\times \frac{I_{corr}\;M}{V\cdot D\cdot A}\times 100\%$$

*I_corr_* is the corrosion current and is determined by an intersection of the linear portions of the anodic and cathodic sections of the Tafel curves, *M* is the atomic mass, *V* is the valence (number of electrons that are lost during the oxidation reaction), *D* is the density, and *A* is the exposed area of the sample.

The corrosion protection efficiency from Tafel polarization curves was calculated by the following formula [44,45]:(2)$$\eta =\frac{J_{corr}\left(Substrate\right)-J_{corr}\left(Coating\right)}{J_{corr}\left(Substrate\right)}\times 100\%$$

In this equation, *J_corr_* (*Substrate*) and *J_corr_* (*Coating*) correspond to the corrosion current densities of bare aluminum and coated aluminum, respectively.

Electrochemical impedance spectroscopy (EIS) (PalmSens BV, Houten, Netherlands) is one of the most efficient and intensively used methods for investigation and prediction of corrosion protection. It is a very sensitive detector and a nondestructive measurement of the condition of a coated metal and it tracks the coated sample as it changes, identifiying the cause of coating failure. Moreover, the EIS spectrum is a relative measurement, so it is used to compare it to another EIS spectrum. 

The EIS measurements were carried out by the potentiostat. On one hand, a sinusoidal AC voltage with a constant amplitude of 0.07 V_RMS_ and a frequency ranging from 100 mHz to 100 KHz is applied between the reference electrode and the working electrode. On the other hand, the current response is analyzed through the counter electrode to extract the phase and amplitude relationship between the current and the voltage signals, acquiring the impedance spectra in the frequency range with 10 points per decade. The electrochemical cell and all the electrodes are placed inside a grounded Faraday Cage (see Figure 1). Once the sample has reached a steady state, signaled by a stable value of Eoc (better than 0.1 mV/s), the EIS experiment can start.

The use of impedance, *Z*, which is a measure of a circuit’s tendency to resist (or impede) the flow of an alternating electrical current, has the following equivalent mathematical expression:(3)$$\begin{array}{c} Z\left(\theta_{v}-\theta_{i}\right)=\frac{V_{R_{e}-W_{e}}\sin\left(\omega t+\theta_{v}\right)}{I_{Ce}\sin\left(\omega t+\theta_{i}\right)}\\ \omega =2\pi f_{n};f_{n}=0.1\mathrm{Hz}\to 100\mathrm{KHz} \end{array}$$

In Equation (3), *V_Re_*_−_*_We_* corresponds to the voltage applied between the the reference electrode and the working electrode, *I_ce_* is counter electrode current, *f* is the frequency, *ω* is the angular frequency, and *θ* is the phase for the different variables. 

Vis-NIR spectroscopy has been used to characterize the optical properties of the PVDF-HFP electrospun SHS and SLIPS. The optical transmittance and reflectance spectra were carried out in the spectral range from 400 nm to 1600 nm on the Vis-NIR spectrophotometer at room temperature. A spectrophotometer from INSTRUMENT SYSTEM (Instrument Systems Optische Messtechnik GmbH, Munich, Germany) has been used to perform all the measurements.

Finally, to evaluate the ice adhesion of the proposed materials, a method inspired in the AMIL design centrifuge adhesion test (CAT) was used [46]. This facility is located in the National Institute of Aerospace technology of Spain (INTA), and both the icing wind tunnel (IWT) and the CAT are inside a 54 m^3^ cold climate chamber, offering temperature stability during the ice accretion, the transport to the ice adhesion rig, and the ice adhesion test. Details of this facility and the methods to control the conditions have been previously reported [47,48]. The procedure has two main parts:(1)Ice accretion on samples: an important advantage of impact icing in IWTs, comparing with the in-mold icing, is the capability to replicate in-flight icing conditions in terms of air speed and temperature, and the generation of clouds of supercooled water where the liquid water content (LWC) and droplet size (MVD: median volume diameter) are controlled (see Figure 2 and Figure 3).

In this study, two icing conditions have been used (see Table 1).

(2)Ice adhesion test: CAT samples are mounted on a rotating beam 17 cm long where the accreted ice is placed in the last 2.5 cm to decrease the differences in the angular speed. Then, the probe is accelerated at a rate of 300 rpm/s until a piezoelectric sensor detects the impact of ice detached, when the corresponding angular speed is taken to calculate the centrifugal force (see Figure 4).

The Ice adhesion obtained from this test is defined as the shear stress needed to detach the ice from the surface, calculated as the ratio between the centrifugal force and area ratio (see Equation (4)).
(4)τ=F/A
where

*τ* is the shear stress (Pa);*F* is the centrifugal force (N);*A* is the Iced area (m^2^).

On the other hand, to calculate the centrifugal force, Equation (5) was used:(5)$$F=mr\omega^{2}$$
where

*m* is the mass of ice (kg);*r* is the radius of the beam (m);*ω* is the angular speed of rotation (rad/s).

Three replicates of each material were consecutively tested until the full coating degradation, as well as three samples of the bare substrate (AW 7075-T6) and three made from PTFE for comparison purposes. The mean value and standard deviation were calculated for all results of every material.

## 3. Results and Discussion

First of all, the influence of applied voltage and flow rate as operational input parameters for a fixed polymeric precursor concentration (15 wt.%) has been deeply analyzed in terms of electrospinning jet modes, thickness control, surface morphology, wettability, water adhesion strength, optical transmittance, anticorrosion, and ice adhesion. The lower- and upper-limit values of the applied voltage and flow rates have been studied to determine a properly electrodeposition process. Finally, five samples were successfully electrospun at the selected applied voltages and flow rates conditions.

### 3.1. Electrospinning Jet Modes 

The jet modes of the electrospinning procedure have been studied to understand the effect of varying these two specific parameters (applied voltage and flow rate), in order to determine the optimum deposition mode. On one hand, one of the main characteristics of the electrospinning procedure is to achieve a stable jet, hence a map of the different jet modes areas has been analyzed (see Figure 5). The lower and upper limit values (denoted as V_min_ and V_max_) of the applied voltage were obtained by varying the voltage at different fixed flow rates (from 500 µL/h to 4000 µL/h). Figure 6 shows how the jet instability is caused outside the limits of the cone jet region, resulting in a dripping jet mode when the applied voltage is lower than the V_min_; on the other side, an oscillating jet mode is achieved when the applied voltage is higher than the V_max_. 

These results are in concordance with the literature, where previous works have demonstrated that the applied field depends on the flow rate and other process parameters such as distance of tip to collector, polymeric precursor properties (concentration, molecular weight, viscosity, surface tension, and nature of solvent), and environmental conditions (relative humidity and temperature). A variety of jets can emerge from the tip of the nozzle [49]. When increasing the potential voltage to the nozzle, the electrospinning mode changes into dripping mode, cone jet mode, and oscillating jet mode, respectively (see Figure 6). In a dripping mode (Figure 6a), spherical or spindle-like droplets elongated in the direction of electric field emerge from the meniscus. The dripping frequency increases as the voltage is increased, until a transition is observed from dripping to a stable jet, due to the equilibrium of the electric forces with the surface tension and gravity [50]. Thus, the dripping mode changes to cone jet mode, and the liquid body acquires a conical shape, with a half angle of 49.3° (a whole angle of 98.6°), referred to as the Taylor Cone [51]. The cone jet mode (Figure 6b) is a steady jet mode, in which case, the supplied liquid forms a regular, axisymmetric cone with a thin and elongated jet that splits into numerous sub-jets. However, the Taylor cone form can lose its axial symmetry while the jet shifts laterally with an elevated potential, such as the case with the oscillating jet mode (Figure 6c), which is caused by the non-axisymmetric or whipping instability. Finally, at even higher voltage, the jet splits up and establishes several emitting sites around the capillary. In the so-called multi-jet mode (Figure 6d), the number of micro jets tends to increase with the applied voltage [52].

Therefore, a set of five samples was successfully electrospun in the cone jet area at different applied voltages and flow rates, respectively.

### 3.2. Thickness Control

The thickness growth of the electrospun samples has been studied to understand the effect of varying two specific parameters (applied voltage and flow rate) and thereby be able to implement a thickness control process. The samples were measured by confocal microscopy in several locations, obtaining a representative number of measurements to determine the average values and at different electrodeposition times (see Figure 7). As expected, the experimental results presented in Figure 7 show a linear relationship between the thickness and the deposition time in samples with the same conditions (voltage and flow rate). Moreover, there is an increment in the slope of the linear trends, when there is a change of the operational point, which means an increment of the applied voltage and flow rate parameters. This increment of the slope has been generated due to an increase in the amount of material deposited, as it can be observed in the Figure 7.

Once the rate of the thickness growth has been identified for the operational parameters, a thickness control system has been developed to obtain samples with 200 µm of thickness for the five operational points (see Table 2). In addition, Table 2 indicates that a transition from P1 to P5 means an increase in the applied voltage and flow rate, involving a decrease of the deposition time.

Finally, in order to prove the final thickness of the developed samples, cross-section confocal images have been used (see Figure 8a). The measurements are defined as the difference in level between the substrate and the top of the coating using the step function of a selected vertical profile (see Figure 8b). The final samples show a thickness of 200 ± 5 µm. 

### 3.3. Surface Morphology 

The final morphology of the electrospun samples has been studied to understand the effect of varying these five operational points. The samples were measured by AFM, FE-SEM, and confocal microscope in several locations to obtain a representative number of measurements and determine the roughness and fiber diameter mean values and their standard deviations, which are represented by the error bars. One of the most important morphology characteristics is the average surface roughness (Sa). Figure 9a shows a similar behavior of Sa in several samples (from S1 to S5). Moreover, the combination of increasing the flow rate and the applied voltage to a greater or lesser degree can increase or decrease the Sa value, obtaining the highest value in the P3 sample. 

The SEM images of the samples (from S1 to S5) presented in Figure 10 have been analyzed through the most representative 50 fibers per sample in order to understand the influence of morphological characteristic such as fiber diameter, fiber density, and fiber size distribution in the values of Sa. In Figure 11, the fiber diameter of the electrospun fibers indicates an overall negative tendency from P1 to P5 due to an increase of the applied voltage and flow rate. Furthermore, the SEM images show a dense structure of entangled threads in all the samples. However, some differences have been observed. On one side, from Figure 10a,g, it is possible to confirm the density increment of the fibers present, which is clearly higher in Figure 10g, because of an increment of flow rate and voltage. On the other side, the lower-surface-roughness images represented in Figure 10a,g, which correspond to the operational points of P1 and P5, exhibit a lower standard deviation, involving a higher homogenization of the fiber size distribution, in spite of having different average fiber diameters (see Figure 11). On the contrary, Figure 10e, which characterizes the operational point of P3, where the surface roughness reaches its higher value, presents a visible increment of the standard deviation (see Figure 11), which means a reduction of the fiber size uniformity. Therefore, the influence of the fiber diameter variability generates a highly rough structure caused by the hierarchical microstructures and nanostructures of micro/nanofibrous, as can be appreciated in Figure 11, where there is a clear correlation between the average surface roughness and the standard deviation of the fiber diameter. 

These results are in concordance with the literature, where previous works have demostrated that a high voltage allows stretching forces capable of promoting the formation of uniformly distributed fibers [53] and the effect of increasing the applied voltage produces narrower fibers due to the production of a higher electrostic repulsive force on the fluid jet [54,55]. Nevertheless, taking into account other works, the adequate selection of the voltage depends on the system used, such as type of solvent and polymer concentration [56,57]. Therefore, the creation of micro/nano multilevel structures through the combination of micro/nanofibrous mat generates a highly rough structure [36].

### 3.4. Wettability Properties

When a water droplet rests on a solid surface, a physicochemical balance is generated. This balance state is driven by the chemistry and the geometrical structure of the surface. On one side, the chemistry surface interacts through three surface free energies (liquid–gas, solid–gas, and solid–liquid), which are explained by the Young’s equation (see Equation (6)).
(6)$$\theta_{Y}=\frac{\gamma_{SG}-\gamma_{LS}}{\gamma_{LG}}$$
where $\gamma_{LG},\gamma_{SG},\gamma_{SL}$ represent the surface free energies (N/m) from liquid–gas, solid–vapor, and solid–liquid, respectively. In addition, $\theta_{Y}$ is the equilibrium water contact angle (EWCA) for a smooth surface [58].

In this case, the PVDF-HFP coating has low values of surface free energies due to the presence of fluoride functional groups [59]. A previous work has shown that this value is around 10 mN·m^−1^ [60], although this surface energy value strongly depends on the content of HFP in the PVDF-co-HFP because it is associated to an increase in the fluorine content in the polymer [61]. In Figure 12, the FTIR spectrum of the PVDF-co-HFP electrospun fibers is shown. A strong absorption peak at 1174 cm^−1^ can be clearly appreciated, which is assigned to the symmetrical stretching mode of –CF2 group, whereas the peak at 1390 cm^−1^ is assigned to the CH2 groups of PVDF-HFP [62].

Thus, smooth PVDF-HFP coatings have shown a hydrophobic behavior. On the other side, the geometrical structure of the surface plays an important role. This is the reason why the Young model, which is applied for smooth surfaces, is a poor model to determine the behavior for a certain surface roughness. Thus, in order to understand the influence of rough surfaces, some different models have been reported. In this work, the Cassie–Baxter model has been selected to explain the wetting surface state due to the presence of air entrapped inside the gaps of the fibers net, which makes the penetration of the liquid into the roughness or texture of the surface difficult due to capillary forces. The Cassie–Baxter equation (see Equation (7)) for solid–liquid–air composite surface (porous) considers the effect of air gaps under the droplets.
(7)$$\theta_{CB}=X_{s}\cos\left(\theta_{Y}+1\right)-1$$
where $X_{s}$ is the solid surface fraction (%), $\theta_{Y}$ is the EWCA for a smooth surface, which is described in Equation (6), and $\theta_{CB}$ is the observed static water contact angle (WCA).

Thus, in order to characterize the wetting surface, the WCA of several samples has been measured. On one hand, the experimental results presented in Figure 13 show a superhydrophobic behavior (WCA > 150°) in all the samples of study, achieving the highest value in P3 (161.99° ± 2.42°). On the other hand, there is a linear relationship between the WCA and the average roughness (Sa). Therefore, the samples with a higher value of the resultant roughness have presented an increase in the wettability.

According to this, the improvement in the wettability properties of the electrospun fibers mats is mainly caused by the multilevel roughness of the surface, which is induced by the combination of micro/nano-sized fibers, which generates a substantial increase in air gaps, leading to a superhydrophobic surface [63,64].

### 3.5. Anticorrosion Performance

In order to check the anticorrosion behavior, Tafel polarization test and electrochemical impedance spectroscopy were performed. Firstly, the aluminum substrate (AA7075T6) was tested as a reference (Figure 14a) to be later compared with the electrospun samples with 200 µm of thickness for the five operational points previously mentioned, obtaining the Figure 14b typology. Secondly, to improve the abrasion resistance, the electrospinning coating was deposited over a glue layer with 60 µm of thickness (see Figure 14c) to bind the electrospun layer with the aluminum substrate. Hence, the influence of the glue layer (Figure 14d) was analyzed under the conditions mentioned above. 

#### 3.5.1. Tafel Polarization Test

The corrosion results of Tafel polarization show that all the electrospun samples minimize the corrosion current density and the corrosion rate of the aluminum substrate in three orders of magnitude, which means a considerable improvement in the resultant protection efficiency (see Table 3). 

Furthermore, Table 3 indicates that an increase of the voltage and flow rate from S1 to S5, which corresponds to the Figure 14b typology sample, causes a small decrease of the corrosion current density (Jcorr) and a lower corrosion rate with an enhancement of the protection efficiency (η). However, the opposite effect was obtained with the influence of the glue layer (see Figure 14d), from S1G to S5G, where an increase of the corrosion current density (Jcorr), a higher corrosion rate, and a loss of the protection efficiency (η) are achieved. In addition, the difference between the aluminum substrate and the substrate with only the glue coating (Figure 14c) implies an improvement of the protection efficiency in one order of magnitude. The best result has been found for the sample S1G, which shows the lowest corrosion current density and corrosion rate in comparison with the other samples.

Finally, the samples with a lower value of average fiber diameter and a greater homogenization fiber size exhibit a higher reduction in the corrosion rate. This phenomenon could be due to a better blocking of the corrosion current because of the atomized distribution of pores through the mesh of thin fibers, which create a barrier to retain the electrolyte infiltration to a larger extent. This effect is more intense for the samples S3, S4, and S5 where the fibers are thinner and the network of pores more atomized and efficient. Furthermore, the samples coated with the glue layer combine the effect of two barriers, and this combination is more protective for the samples with the thicker fibers where the pores are broader and the glue has impregnated a thicker part of the mesh. Finally, the results of Tafel polarization curves are presented in Figure 15.

#### 3.5.2. Electrochemical Impedance Spectroscopy (EIS)

In the present study, the Sx and SxG electrospun coated samples were exposed to an electrolyte (6 wt% NaCl in water) at room temperature for an immersion time of 24 h and 72 h, respectively. The EIS results can be seen in the Bode diagrams of Figure 16 and the Nyquist plots shown in Figure 17. The Bode diagrams for the samples without glue interlayer (S1–S5) show a clear tendence in the impedance at low frequency that is perfectly correlated with the corrosion currents and rates listed in Table 3: the higher the corrosion rate, the lower the impedance. The same is observed for the samples with the glue interlayer (S1G–S5G).

On the other hand, the Nyquist plots allow to distinguish the different behavior of some coatings: with the exception of the samples that show better corrosion behavior (S5 and S1G), a second time constant emerges for all the others and a second semicircle becomes more visible as the coating becomes less protective because the porous mesh becomes more permeable (samples S4 → S3 → S2 → S1) or the glue infiltrated in this mesh reduces its effective thickness (samples S2G → S3G → S4G → S5G).

Considering these results, an adjustment with the typical equivalent circuit is required for a coating with defects i.e., a double-loop circuit, except for the S5 and S1G samples, which fit better with a single-loop circuit (see Figure 18).

First of all, the value of R_s_ is the resistance of the electrolyte solution between the working electrode and reference electrode. In this case, the electrolyte is very conductive, so R_s_ is very low (18–22 Ω) for all the samples (see Table 4).

The R_pore_ indicates the resistance of the coating due to the penetration of electrolyte into the micropores of the coating. Thus, if the barrier properties of the coating are good, a higher value of R_pore_ is obtained. Moreover, the R_ct_ depends on the fraction of the substrate area that is in contact with the electrolyte. Therefore, a smaller exposed area results in a higher R_ct_ and thereby a lower corrosion rate. In Table 4, the samples (S1 → S5) and (S5G → S1G) exhibit a clear tendency: the lower the corrosion rate, the higher R_pore_ and R_ct_.

Finally, the constant phase element (CPE) describes an “imperfect capacitor” of the coating and aluminum electrode due to several factors in the diffusion process: the surface heterogeneity, electrode porosity, roughness or non-uniform current, and potential distribution. It also fits the results better than an ordinary capacitor does and is mathematically expressed as: (8)$$Z_{CPE_{i}}=\frac{1}{Q_{0_{i}}{\left(j\omega \right)}^{n_{i}}};C_{CPE_{i}}=\frac{{\left(Q_{0_{i}}\cdot R_{i}\right)}^{\frac{1}{n_{i}}}}{R_{i}};{\left\{\begin{array}{c} Z_{CPE_{i}}\\ R_{i} \end{array}\right\}}_{i}={\left\{\begin{array}{c} Z_{CPE_{coating}}\\ R_{pore} \end{array}\right\}}_{i=1};{\left\{\begin{array}{c} Z_{CPE_{dl}}\\ R_{ct} \end{array}\right\}}_{i=2}$$
where $Z_{CPE_{i}}$ is the CPE impedance ($Z_{CPE_{coating}}$, $Z_{CPE_{dl}}$), $C_{CPE_{i}}$ is the CPE capacitance ($C_{CPE_{coating}}$, $C_{CPE_{dl}}$), $R_{i}$ is the parallel resistance ($R_{pore}$,$R_{ct}$), $Q_{0_{i}}$ is a constant, *j* = (−1)^½^, *ω* = 2π*f*, *f* is the frequency, $n_{i}$ is a constant (between 0 and 1), and the subindex *i* stands for the number of the loop: I = 1 (RC coating elements) or I = 2 (RC electrolyte–electrode interface elements).

On one hand, the *C_CPE coating_* parameter tends to be rather low (see Table 4), and there is no clear correlation between the samples. By contrast, the *C_CPE dl_* values are much higher than *C_CPE coating_*, and a trend can be observed (see Table 4). The samples (S1 → S5) and (S5G → S1G) exhibit a decrease of the *C_CPE dl_*, which corresponds to an increment of the corrosion protection.

Thereby, the EIS results corroborate the noticed behavior of Tafel polarization test, which contributes to a better understanding of the corrosion protection effect.

### 3.6. SLIPS Wetting Properties

In order to develop SLIPS, there are three criteria to be followed. Firstly, porous microstructures and a micro/nanoscale structure template are necessary to lock the lubricating liquid. Secondly, it should be easier to get the solid substrate wetted with lubricating liquid than with other liquids. Thirdly, the lubricating and impinging test liquids must be immiscible. Such surfaces show ultralow contact angle hysteresis as well as enhanced liquid repellency [28].

Thus, to reduce the adhesion forces, a lubricating film was applied. The silicon oil was used as lubricant for SLIPS, filling the grooves of the electrospun porous surface and increasing the capillary forces. However, a decrease of the WCA was generated due to a reduction of roughness [65], as shown in Figure 19.

Although the S3G-SLIPS has lower static contact angles, it also has significantly lower contact angle hysteresis than the as-produced PVDF-HFP electrospun coating does. Therefore, despite lower contact angle of SLIPS, a water droplet would have much higher mobility on the SLIPS compared with lubricant-free structure. This is an expected effect of the lubricants reported elsewhere [66].

Finally, the sliding water angle ($\alpha_{slide}$) and the contact angle of hysteresis (CAH) of the S3G-SLIPS and S3G samples (see in Figure 20) obtained a higher performance through lower CAH and $\alpha_{slide}$ values. This implies a low water adhesion strength generated by the infused silicon oil. 

As a consequence, the SLIPS has visible slippery properties, such as drops or stains not remaining on or attaching to the surface, due to the rapid sliding movement of the water produced by the fluid interface with the coating. The other main characteristic of SLIPS is the self-healing. If damage is produced on the surface, it is quickly repaired by the lubricant flow that covers the damage zone [17].

### 3.7. Optical Properties

The optical properties are an interesting research point because a visible change of the sample to totally transparent was observed when the sample was infused with silicon oil. For this reason, Vis-NIR spectra of the SHS and SLIPS samples have been analyzed in the spectra range of 400–1600 nm in terms of the transmittance and reflectance, as can be appreciated in Figure 21.

The main conclusion derived from these experimental results is associated to a homogenization process of the refractive index due to a lubricating layer [65]. On one hand, this effect implies an improvement of the light transmittance from 10% in the SHS coating up to a 90% in SLIPS (see Figure 21a). On the other hand, the SHS coating shows a reflectance of 90% (see Figure 21b), where the scattering of the light plays an important role, generated by the air pockets, which are trapped between the protrusions on the fiber surface. Therefore, if the silicon oil fills the gaps of the electrospun porous surface, a loss of the scattering phenomena is produced and consequently, there is a decrease up to 10% of the light reflectance. This kind of optical measurement can be useful in order to implement fast testing methods in practical applications.

### 3.8. Ice Adhesion Performance

The two proposed materials have presented a very different behavior in two icing atmospheric representative conditions (glaze and rime). While the lubricated solution achieved very low ice adhesion for the two ice types, the nonlubricated results were worse than those of the PTFE surfaces (Figure 22).

To represent the range of improvement of an anti-icing solution with respect to the bare substrate, the adhesion reduction factor (ARF) has been used in the bibliography [46], which is calculated by the ratio of the ice adhesion of the bare substrate and the proposed solution.

As previously reported [67,68], the lubricated coatings led to low ice adhesion, reducing the ice adhesion 12 times with respect to the uncoated aluminum, and even 4 times with respect to PTFE (see Figure 23), a well-known low ice adhesion material [68,69]. 

Glaze and rime ice have different icing mechanisms [70], but the lubricated coating achieved good results in both. Relevant differences in ice adhesion values are usually found for different methodologies [71], and even in a same methodology, there are important differences in function of the type of ice, the droplet size, or velocity of impact [68,72]. Glaze and rime results reasonably matched with those obtained by Stenroos for aluminum and PTFE, in similar conditions [73]. In our case, the differences are lower, probably due to the more similar droplet size used in both conditions (20 and 40 µm).

However, the durability is still an issue in anti-icing materials [74,75]. In order to address the durability of the coatings, the samples were exposed repetitively to the entire accretion–detach process with the intention of evaluating how many of these cycles can be withstood before degradation. The nonlubricated material is considered to have failed after just one adhesion test, because the material teared during the CAT test, achieving high and dispersed adhesion values. The lubricated one, on the other hand, kept its good behavior until the complete release of the coating after four cycles of accretion/detachment. It is worth noting that aluminum and PTFE samples did not show any sign of degradation. Thus, the durability of the coatings must be improved.

## 4. Conclusions

In the present study, five operational points in the “Cone jet” electrospinning mode area have been correctly identified by controlling the flow rate and applied voltage, as input parameters, for a fixed PVDF-HFP polymeric concentration.

In addition, a thickness control system was successfully implemented to obtain electrospun coatings with 200 µm of thickness at the desired operational parameters. The resultant samples, which were analyzed by confocal microscope, indicate a linear relationship between the thickness and the deposition time in samples with the same conditions.

The surface morphology of the samples analyzed by AFM, SEM, and confocal microscope shows a density increment of entangled threads and an overall negative tendency of the fiber diameter caused by an increase of both input parameters. Finally, an increment of the surface roughness as a result of the hierarchical microstructures and nanostructures of micro/nanofibers is achieved, where there is a reduction of the fiber size uniformity.

In this way, the experimental results corroborate the important role of obtaining hierarchical nano/microfibrous composite and the creation of micro/nano multilevel structures for the optimization of superhydrophobic surfaces, showing a WCA higher than 150°. In addition, the corrosion resistance exhibits an excellent performance, and some differences have been noticed. On one hand, a lower value of average fiber diameter and a greater homogenization fiber size show a higher reduction in the corrosion rate due to a better blocking of the corrosion current by the mesh of thin fibers and the atomization of the air pockets. On the other hand, the performance of the coatings deposited on the glue interlayer results in the opposite effect, due to the impregnation of the mesh of fibers by the glue, generating the effect of two barriers. This combination is more protective for the samples with the thicker fibers where the pores are broader and the glue has impregnated a thicker part of the mesh.

EIS analysis corroborates this explanation: the thinner the fiber diameter, the higher the R_pore_ for the unglued coatings, and just the opposite for the coatings deposited on the glue interlayer.

A SLIPS was obtained, filling with the micro/nanoscale structure with lubricating oil instead of air in the SHS. On one side, the WCA shows a hydrophobic behavior generated by a decrease of roughness, and on the other side, the CAH and transmittance results exhibit low water adhesion strength and excellent optical transparency. In addition, the SLIPS exhibits four and seven times lower ice adhesion with respect to PTFE in both analyzed icing atmospheric conditions, glaze and rime, respectively. This means that the lubricated coating reduces the ice adhesion 12 times with respect to the uncoated aluminum, meaning an improvement.

Finally, we have demonstrated a scalable and reproducible coating method to create a slippery, icephobic surface on aluminum substrate. Compared to conventional materials, this presents a great opportunity to utilize SLIPS-based icephobic surfaces for aerospace applications, as well as in other high-humidity environments, when icephobic surfaces are desirable.

## Figures and Tables

**Figure 1 polymers-13-04164-f001:**
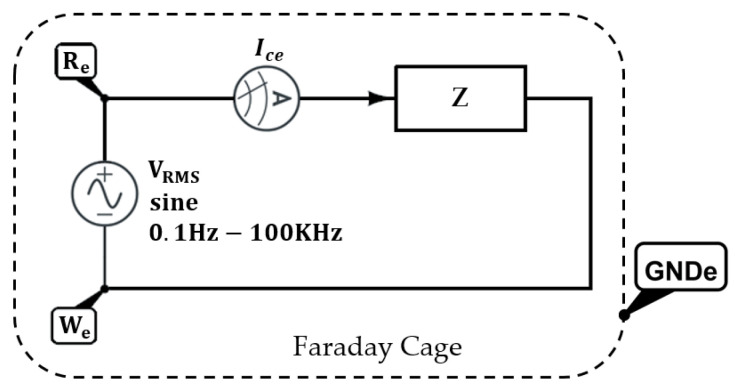
EIS circuit setup. Counter electrode current (*I_ce_*), reference electrode (*R_e_*), working electrode (*W_e_*), ground electrode (GNDe), and circuit impedance (*Z*).

**Figure 2 polymers-13-04164-f002:**
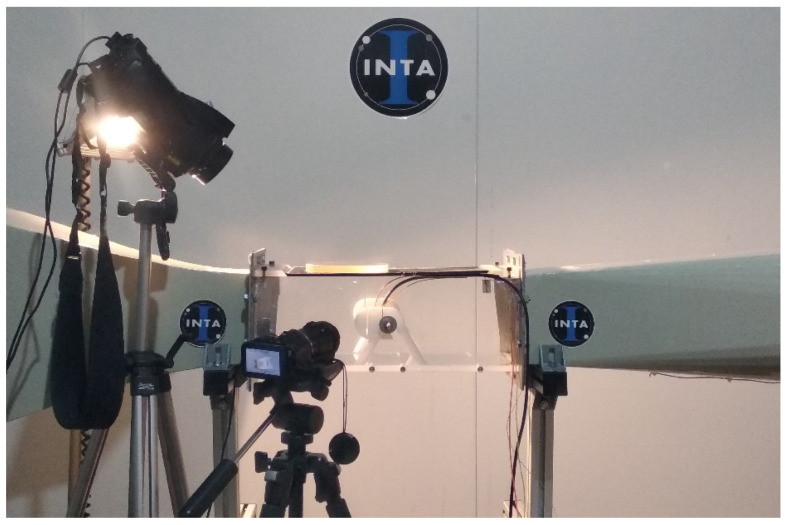
Test section of the icing wind tunnel (IWT) in a cold climate chamber at INTA, Spain.

**Figure 3 polymers-13-04164-f003:**
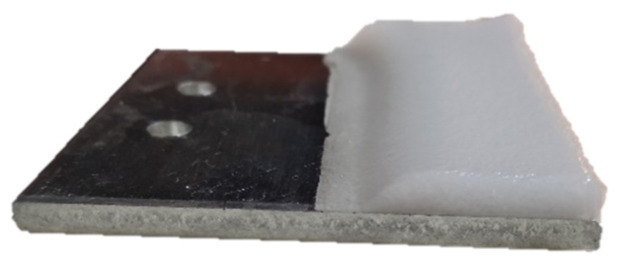
The ice was accreted in one half of the area of the 50 × 50 mm^2^ samples using a mask to have an iced area of 12.5 cm^2^.

**Figure 4 polymers-13-04164-f004:**
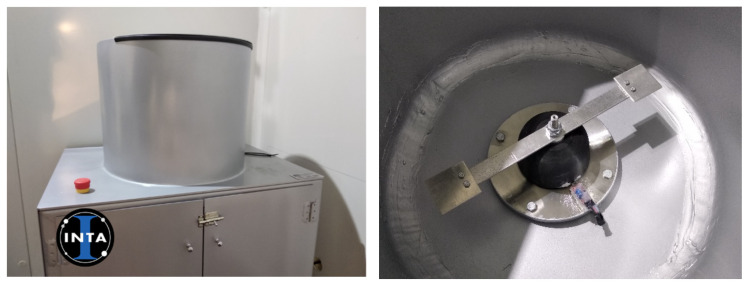
Centrifuge adhesion test (CAT) on a rotatory beam.

**Figure 5 polymers-13-04164-f005:**
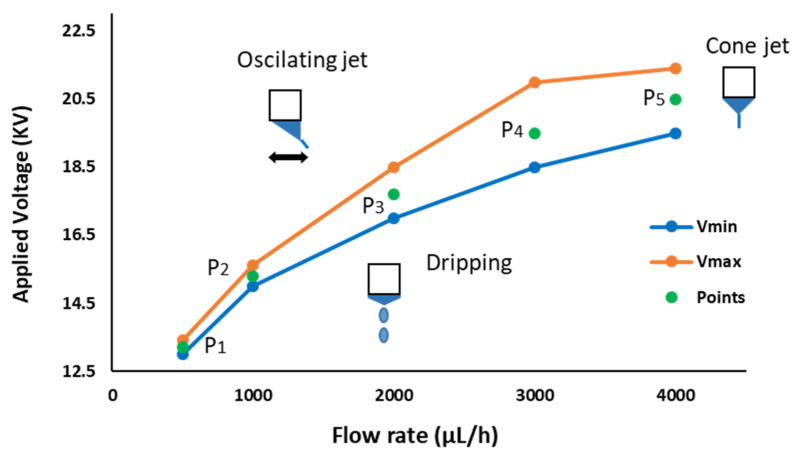
Operating diagram for PVDF-HFP, electrospinning jet modes: the dripping region delimited with the cone jet region by V_min_ and the oscillating jet region delimited with the cone jet region by V_max_. Within the cone jet area, there are five selected samples-points to study: P1 (13.2 KV; 500 µL/h), P2 (15.5 KV; 1000 µL/h), P3 (17.7 KV; 2000 µL/h), P4 (19.5 KV; 3000 µL/h), and P5 (20.5 KV; 4000 µL/h).

**Figure 6 polymers-13-04164-f006:**
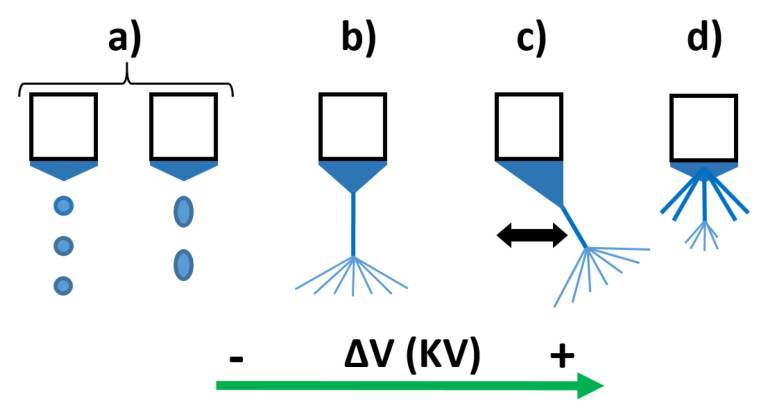
Schematic diagram of main electrospinning modes: (**a**) dripping mode, (**b**) cone jet mode, (**c**) oscillating mode, and (**d**) multi-jet mode.

**Figure 7 polymers-13-04164-f007:**
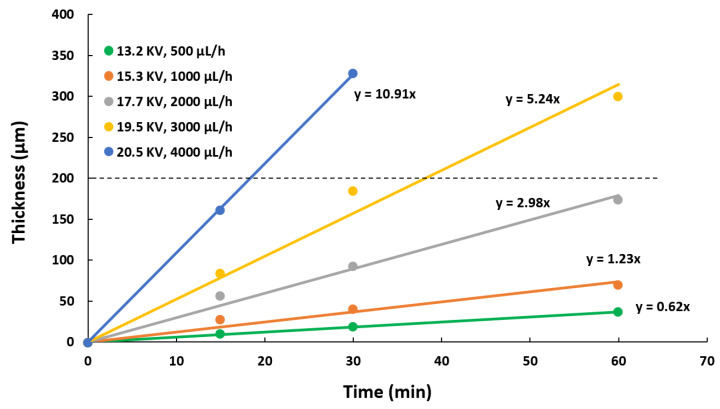
Evolution of the thickness as a function of the deposition time for different operating conditions of flow rate and applied voltage.

**Figure 8 polymers-13-04164-f008:**
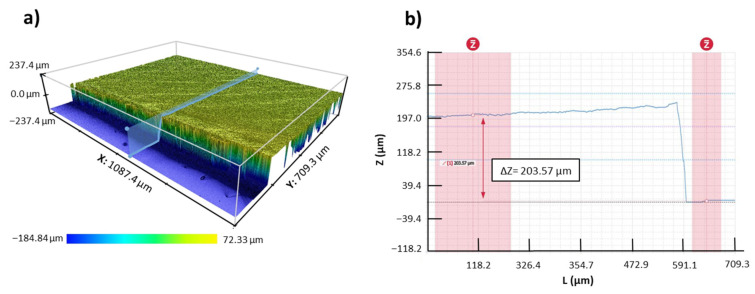
(**a**) 3D confocal image of the cross-section S3 sample, where a restore algorithm and a vertical profile have been selected. (**b**) 2D image of the vertical profile with the step function measurement.

**Figure 9 polymers-13-04164-f009:**
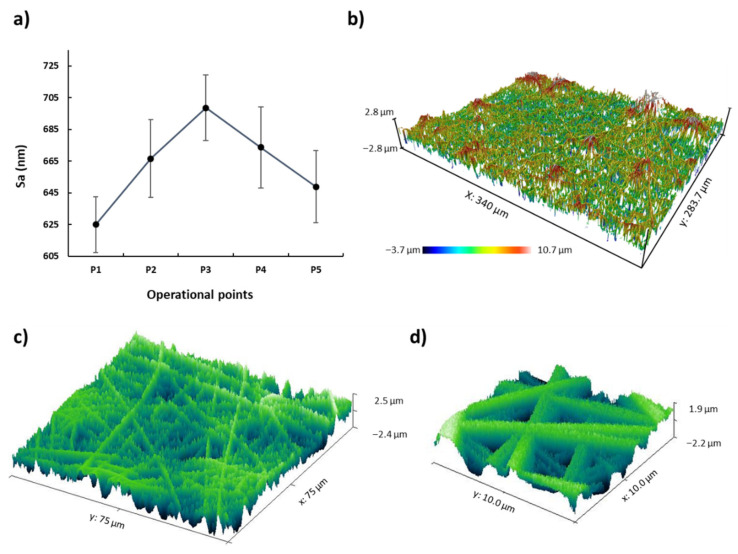
(**a**) Evolution of the average surface roughness (Sa) as a function of the operational points (P1, P2, P3, P4, and P5). 3D images representing the surface morphology of the electrospun mats in sample S3 (2000 µL/h and 17.7 KV) with the (**b**) confocal microscope and AFM (**c**,**d**) in the scale of 75 µm and 10 µm, respectively.

**Figure 10 polymers-13-04164-f010:**
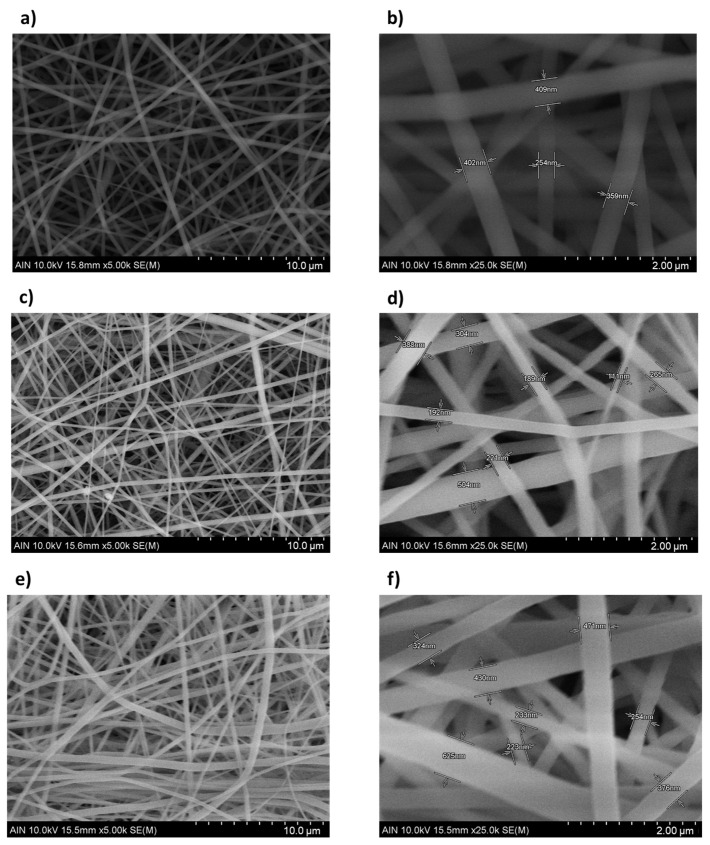
Scanning electron microscopy (SEM) images representing the surface morphology of the electrospun mats for sample S1 (**a**,**b**) (500 µL/h and 13.2 KV), S2 (**c**,**d**) (1000 µL/h and 15.3 KV), S3 (**e**,**f**) (2000 µL/h and 17.7 KV), S4 (**g**,**h**) (3000 µL/h and 19.5 KV), and sample S5 (**i**,**j**) (5000 µL/h and 20.5 KV) at the scale of 10 µm (**a**,**c**,**e**,**g**,**i**) and 2 µm (**b**,**d**,**f**,**h**,**j**), respectively.

**Figure 11 polymers-13-04164-f011:**
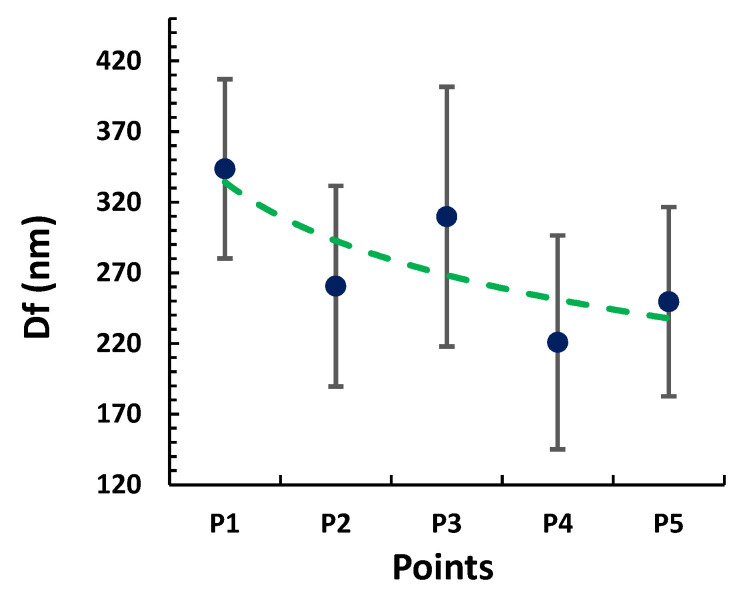
Evolution of the fiber diameter (Df) and its standard deviation as a function of the operational points (P1, P2, P3, P4, and P5).

**Figure 12 polymers-13-04164-f012:**
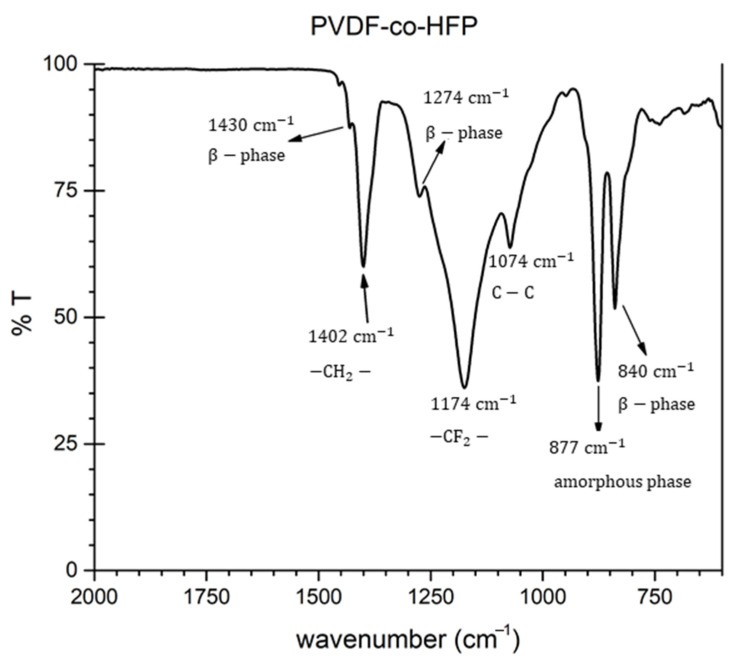
FTIR spectrum of the PVDF-co-HFP electrospun fibers.

**Figure 13 polymers-13-04164-f013:**
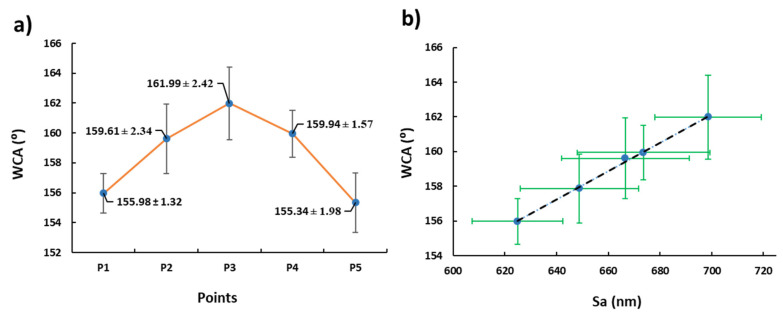
(**a**) Evolution of the water contact angle (WCA) and its standard deviation as a function of the operational points (P1, P2, P3, P4, and P5). (**b**) Behavior of the WCA as a function of the average surface roughness (Sa).

**Figure 14 polymers-13-04164-f014:**
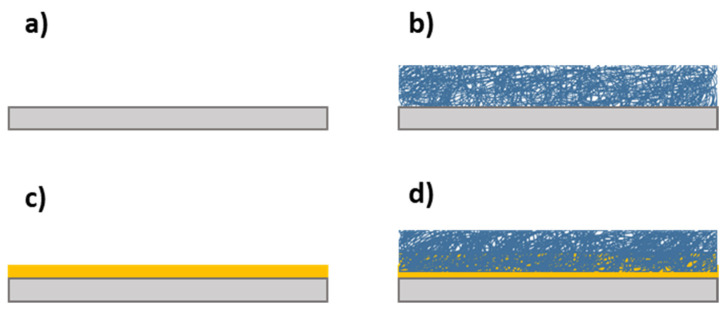
(**a**) Aluminum alloy substrates (AA7075-T6). (**b**) Aluminum substrate and electrospun layer. (**c**) Aluminum substrate with glue layer. (**d**) Aluminum substrate with glue layer and electrospun layer.

**Figure 15 polymers-13-04164-f015:**
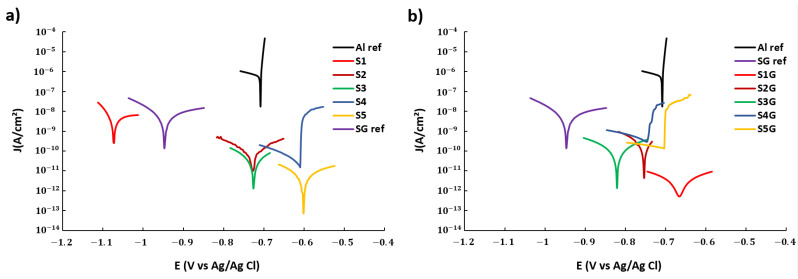
Tafel plots corresponding to the aluminum bare substrate and the different aluminum samples composed of PVDF-HFP electrospun fibers mats without (**a**) and with (**b**) glue layer after being tested in 6 wt% NaCl aqueous solution. For greater clarity, only the interval ±100 mV is shown in each curve.

**Figure 16 polymers-13-04164-f016:**
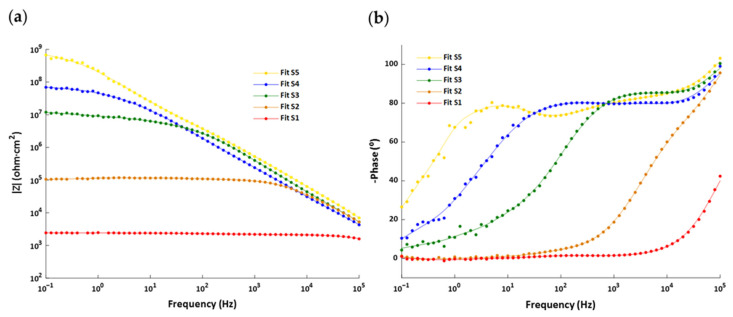
Bode diagrams corresponding to the S1–S5 samples: (**a**) modulus and (**b**) frequencies as well as those corresponding to the S1G–S5G samples: (**c**) modulus and (**d**) frequencies.

**Figure 17 polymers-13-04164-f017:**
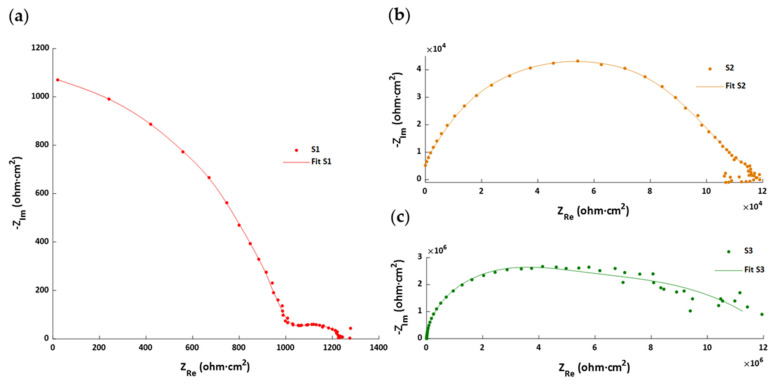
Nyquist plots corresponding to the samples (**a**) S1, (**b**) S2, (**c**) S3, (**d**) S4, (**e**) S5, (**f**) S1G, (**g**) S2G, (**h**) S3G, (**i**) S4G, and (**j**) S5G.

**Figure 18 polymers-13-04164-f018:**
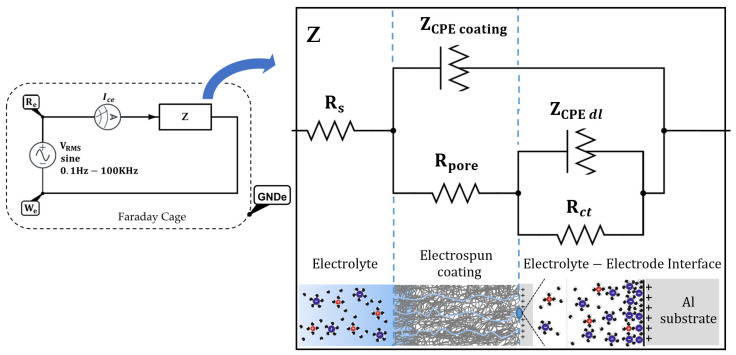
Diagram of the equivalent circuit for a damaged coating. Counter electrode current (*I_ce_*), reference electrode (*R_e_*), working electrode (*W_e_*), ground electrode (GND_e_), circuit impedance (*Z*), resistance of the electrolyte solution (*R_s_*), the pore resistance (*R_pore_*), the charge transfer resistance (*R_ct_*), the constant phase elements of the coating (*Z_CPE coating_*), and the constant phase elements of the doble layer (*Z_CPE dl_*).

**Figure 19 polymers-13-04164-f019:**
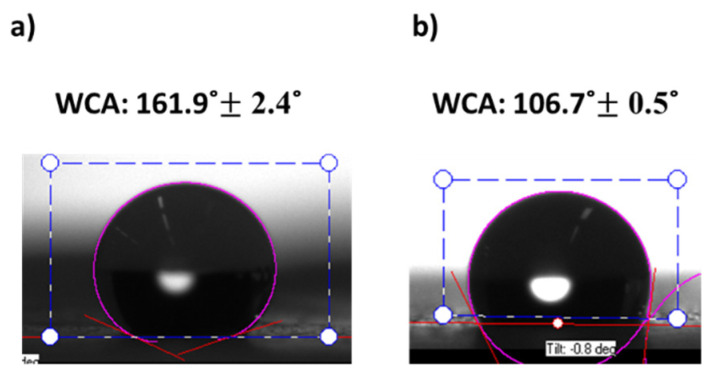
Static water contact angles (WCA) of samples S3G (**a**) and S3G-SLIPS (**b**). The sample S3G (**a**) exhibits a superhydrophobic behavior (WCA < 150°) and samples S3G-SLIPS (**b**) a hydrophobic behavior (90° < WCA < 150°).

**Figure 20 polymers-13-04164-f020:**
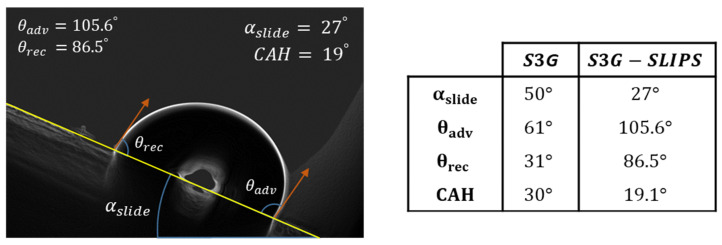
The sliding water angles ($\alpha_{slide}$), the advancing contact angle ($\theta_{adv}$), the receding contact angle ($\theta_{rec}$), and the contact angle of hysteresis (CAH) of the S3G and S3G-SLIPS samples.

**Figure 21 polymers-13-04164-f021:**
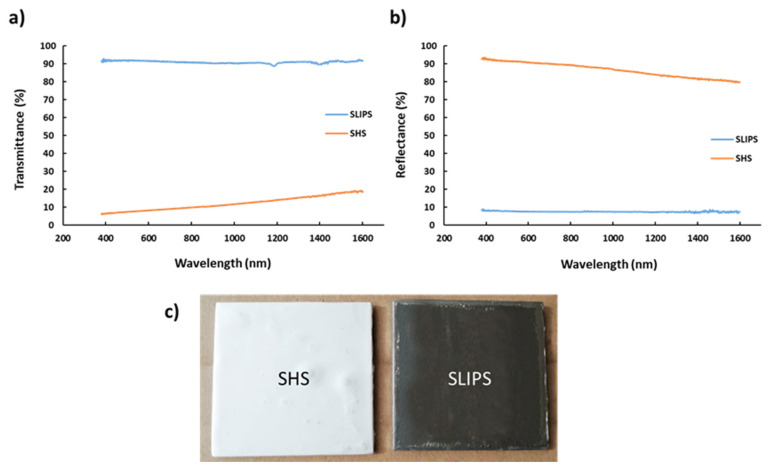
(**a**) Transmittance Vis-NIR spectra of the SHS and SLIPS samples. (**b**) Reflectance Vis-NIR spectra of the SHS and SLIPS samples. (**c**) The resultant SHS and SLIPS appearance.

**Figure 22 polymers-13-04164-f022:**
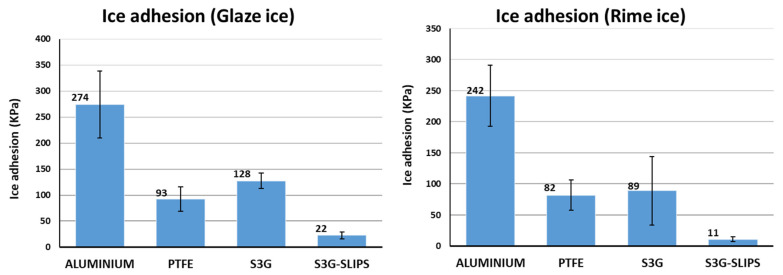
Graphs of ice adhesion tests after glaze and rime icing procedures in IWT.

**Figure 23 polymers-13-04164-f023:**
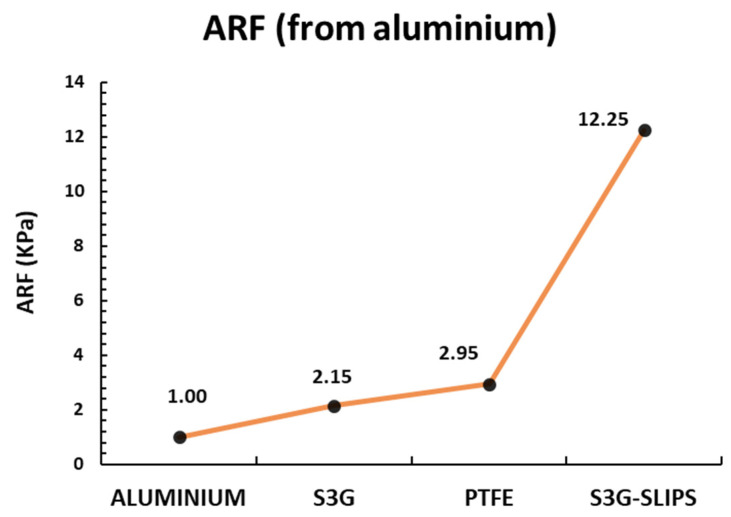
Graphs of the adhesion reduction factor of several coatings: aluminum substrate, PTFE, S3G, and S3G-SLIPS.

**Table 1 polymers-13-04164-t001:** Summary table of the icing test conditions. Condition 1 generates glaze ice, and condition 2 generates rime ice.

Conditions	Condition 1	Condition 2
Wind speed (m/s)	70	70
Temperature (°C)	−5	−15
MVD (µm)	40	20
LWC (g/m^3^)	0.5	0.5
Time of nebulization (min.)	4	4
Iced area (cm^2^)	12.5	12.5

**Table 2 polymers-13-04164-t002:** Summary of the inputs parameters for the fabrication operational points (P_x_) of the electrospun coatings (Sx and SxG) as a function of the flow rate, applied voltage, and deposition time, respectively.

Points (Px)	Samples (Sx/SxG)	Applied Voltage (KV)	Flow Rate (µL/h)	Thickness (µm)	Time (min)
P1	S1/S1G	13.2	500	200 ± 5	325
P2	S2/S2G	15.3	1000	200 ± 5	163
P3	S3/S3G	17.7	2000	200 ± 5	67
P4	S4/S4G	19.5	3000	200 ± 5	38
P5	S5/S5G	20.5	4000	200 ± 5	18

**Table 3 polymers-13-04164-t003:** Summary table of the Tafel analysis for uncoated aluminum substrate (7075-T6) and coated PVDF-HFP electrospun coatings, which correspond to Figure 14b (Sx) and Figure 14d (SxG). All the samples of this study have been tested in 6 wt% NaCl aqueous solution.

Sample	Voltage (KV)	Flow Rate (µL/h)	Jcorr (nA/cm^2^)	Ecorr (V)	Corrosion Rate (µm/year)	βa (V/dec)	βc (V/dec)	η (%)
Al ref	0	0	504.210	−0.71	15,902.00	0.01	0.26	0
S1	13.2	500	29.814	−1.07	940.29	0.15	0.09	94.087
S2	15.3	1000	1.182	−0.73	37.28	1.15	0.94	99.766
S3	17.7	2000	0.142	−0.73	4.46	0.44	0.24	99.972
S4	19.5	3000	0.059	−0.61	1.86	0.01	0.18	99.988
S5	20.5	5000	0.039	−0.60	1.24	1.28	0.39	99.992
SG ref	0	0	34.415	−0.95	1085.40	0.82	0.22	93.174
S1G	13.2	500	0.038	−0.67	1.19	1.97	1.25	99.992
S2G	15.3	1000	0.077	−0.75	2.42	0.01	0.05	99.985
S3G	17.7	2000	0.239	−0.82	7.54	0.28	0.22	99.953
S4G	19.5	3000	0.297	−0.75	9.38	0.07	0.13	99.941
S5G	20.5	5000	0.399	−0.70	12.57	0.11	0.50	99.921

**Table 4 polymers-13-04164-t004:** Summary table of the equivalent circuit elements values, which correspond to Figure 14b (Sx) and Figure 14d (SxG) samples at different immersion times of 24 h and 72 h, respectively. All the samples of this study have been tested in 6 wt% NaCl aqueous solution at room temperature.

Sample	Ru (Ω·cm^2^)	R_pore_ (MΩ·cm^2^)	Z_CPE_ (Coating)			R_ct_ (MΩ·cm^2^)	Z_CPE_ (Doble Layer)			X^2^
			Q_0_ (nMho·S^n^)	*n*	Ccoating (nF·cm^2^)		Q_0_ (nMho·S^n^)	*n*	Cdl (nF·cm^2^)	
S1	20.6	0.0025	1.72	0.95	0.86	0.0002	2208.9	0.76	206.01	0.0029
S2	18	0.1	1.59	0.88	0.49	0.011	258.78	0.86	96.82	0.00025
S3	20.8	4.25	0.59	0.95	0.44	8.28	33.16	0.56	12.13	0.0037
S4	21.3	47.01	1.68	0.89	1.23	28.09	15.26	0.77	11.85	0.0011
S5	19.7	776.2	0.93	0.88	0.89	---	---	---	---	0.0068
S1G	18.9	5270	0.41	0.96	0.42	---	---	---	---	0.0082
S2G	19.3	90	0.34	0.99	0.33	305	0.8	0.76	0.51	0.0017
S3G	20.2	4.81	0.32	1	0.31	1.39	5.53	0.75	1.09	0.0012
S4G	21.8	0.04	0.47	1	0.47	0.055	101.54	0.75	18.12	0.0011
S5G	18.6	0.01	0.57	1	0.57	0.012	1106.8	0.57	43.06	0.00096

## Data Availability

Not applicable.

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
