# Peer review of "Icephobic and Anticorrosion Coatings Deposited by Electrospinning on Aluminum Alloys for Aerospace Applications"

_polymers, 2021, doi:10.3390/polym13234164_

Round 1

Reviewer 1 Report

see attachment

Reviewer 2 Report

To improve the manuscript, please provide additional information and perform the following corrections:

  1. On page 3, in 2.1. Materials and reagents should be specified the producer of AA7075-T6 aluminum alloy;
  2. On page 3, in 2.2. Electrospinning procedure should be specified the silicon oil grade and its producer;
  3. On page 6, in Fig. 3 caption, 50x50 mm should be written as 50 mm x 50 mm or 50 x 50 mm2;
  4. All equations should be numbered and referenced (i.e. on page 7);
  5. On page 8, in the phrase “These results are in concordance with the literature, where previous works have demonstrated that depending on the applied field, flow rate and other process parameters, a variety of jets can emerge from the tip of the nozzle [49].” should be specified the process parameters instead of expressing them in a general way as “other process parameters”;
  6. On page 9, in the legend of Figure 7, the measurement unit of the flow rate “uL/h” should be written as “µL/h”;
  7. On page 12, Figure 10, should be also presented the SEM micrographs for all samples (S1 to S5) and the discussions resulted from the SEM analysis should consider all the developed samples not only S1 and S3 samples; Also, the measurement unit in the scale bar of the SEM micrographs should be “µm” instead of “um”; In Figure 10 should be also presented cross-section SEM images to prove the obtained thickness of 200 µm of the S1-S5 electrospun mats deposited on AA7075-T6 aluminum alloy substrate;
  8. On page 12, Young’s equation should be presented and referenced;
  9. On page 12, referring to the phrase “In this case the PVDF-HFP coating has low values of interfacial free energies due to the presence of fluoride functional groups [58]” should be specified the values of interfacial free energies and the presence of fluoride functional groups in the PVDF-HFP coatings developed in your study should be proved by Fourier-transform infrared (FTIR) spectroscopy;
  10. On page 13, should be clarified the thickness of the glue layer;
  11. On page 15, in Figure 13, On the Oy scale, should be deleted “1 x” that is in front of 10-y (y = 3…14) because 1 x N = N (N is a number ≠ 0);
  12. On page 15, the phrase “In order to develop a slippery liquid-infused porous surface (SLIPS)” is unfinished and the abbreviation for SLIPS was already defined in Abstract, and in page 2; However, each abbreviation should be defined only once in the manuscript, in the first place where is used;
  13. The average surface roughness (Sa) of the S1G to S5G samples should be also specified to prove the claim “However, a decrease of the WCA is generated due to a reduction of roughness [61], as shown in Figure 14.”;
  14. Clarify why the SLIPS wetting properties and ice adhesion performance were studied only for S3G and S3G-SLIPS samples and not for all the developed samples, especially for the sample exhibiting the lowest corrosion rate S1G (1.17 µm/year), and accordingly S1G-SLIPS;
  15. On page 16, in the phrase “For this reason, UV-vis spectra of the SHS and SLIPS samples have been analyzed in the spectral range of 400-1600 nm….” should be replaced “UV-vis spectra” with “Vis-NIR spectra” by considering the mentioned wavelength range; Also, on page 5 should be modified the wavelength values from the text “the spectral range from 300 nm to 1600 nm” to “the spectral range from 400 nm to 1600 nm” (see also the wavelength range from Figure 16);
  16. In the caption of “Figure 16. (a) Transmittance UV-Vis-NIR spectra of the SHS and SLIPS samples (b) Reflectance UV-Vis-NIR spectra of the SHS and SLIPS samples.” should be corrected UV-Vis-NIR with Vis-NIR since in Fig. 16 there is not noticed any recorded spectra in the UV wavelength range of 100-400 nm;
  17. On page 17, in Figure 16. (a) and (b) should be deleted each % that is written near each number from the Oy scale since the measurement unit (%) already appears near the optical property name like Transmittance (%) and Reflectance (%);
  18. On page 17, in “The two proposed materials presented have presented …” correct “presented have presented”;
  19. Referring to the last phrase from the Conclusions “Compared to conventional materials presents a great opportunity to utilize SLIPS-based icephobic surfaces for aerospace applications, as well as in other high-humidity environments, when icephobic surfaces are desirable.” please clarify the maximum sizes of the icephobic surfaces for aerospace applications that could be obtained by scaling-up the sizes of the coated samples and the volume of the PVDF-HFP solution since the Al alloy substrate used in your study had a rectangular parallelepiped shape with 75 mm in length, 50 mm in width and 2 mm in thickness, and the electrospinning procedure was carried out by loading the PVDF-HFP solution into a 10 mL syringe.

Round 2

Reviewer 1 Report

see attachment

Reviewer 2 Report

The performed revision is satisfactory. However, Fig. 1 from this manuscript is identical to Fig. 1 from the article "The Role of the Fiber/Bead Hierarchical Microstructure on the Properties of PVDF Coatings Deposited by Electrospinning" published by the authors in Polymers 13, no. 3: 464. https://doi.org/10.3390/polym13030464 (https://www.mdpi.com/2073-4360/13/3/464) that is also given as reference [33] in this manuscript. Therefore, it is recommended to delete Fig. 1 from the novel manuscript and to mention that the setup for the Tafel polarization test and Electrochemical impedance spectroscopy is presented elsewhere [33]. Accordingly, all the other figures should be renumbered.

Round 3

Reviewer 1 Report

As for the corrections requested on my part, the authors have improved the article in a satisfactory manner, so I propose to publish this article in corrected form.